# Optimization of Fermentation Conditions and Product Identification of a Saponin-Producing Endophytic Fungus

**DOI:** 10.3390/microorganisms11092331

**Published:** 2023-09-16

**Authors:** Qiqi Chen, Jingying Wang, Yuhang Gao, Xiujun Gao, Peisheng Yan

**Affiliations:** 1School of Environment, Harbin Institute of Technology, Harbin 150090, China; 19b929072@stu.hit.edu.cn (Q.C.);; 2School of Marine Science and Technology, Harbin Institute of Technology, Weihai 264209, China

**Keywords:** *Panax ginseng*, fungal endophytes, total saponins, ginsenosides, uniform design, optimization of fermentation conditions, identification of products

## Abstract

Background: Some fungal endophytes isolated from *P. ginseng* may present a new method of obtaining saponins. This experiment aimed to optimize the total saponin yield produced through in vitro fermentation by an endophytic fungus and analyze its saponin species in the fermented extract. Methods: Fermentation protocols were optimized with a uniform design and verified through regression analysis to maximize the total saponin yield. The saponin types under optimal fermentation conditions were then identified and analyzed using Liquid Chromatography–Mass Spectrometry. Results: The *Trametes versicolor* strain NSJ105 (gene accession number: OR144428) isolated from wild ginseng could produce total saponins. The total saponin yield could be increased more than two-fold through the optimization of fermentation conditions. The concentration of the total saponins achieved by the verified protocol 105-DP was close to the predicted value. The fermentation conditions of the 105-DP protocol were as follows: potato concentration 97.3 mg/mL, glucose concentration 20.6 mg/mL, inoculum volume 2.1%, fermentation broth pH 2.1, fermentation temperature 29.2 °C, and fermentation time 6 d. It was detected and analyzed that the fermented extract of 105-DP contained the ginsenosides Rf and Rb_3_. Conclusion: The endophytic fungus *Trametes versicolor* strain NSJ105 has potential application value in saponin production.

## 1. Introduction

As society develops, people are increasingly pursuing a high quality of life. Therefore, they are paying increased attention to tonics with health benefits. *Panax ginseng* (*P. ginseng*) plants include Asian ginseng (*Panax ginseng* M.) and American ginseng (*Panax quinquefolius* L.). As a traditional precious tonic, *P. ginseng* is favored and highly respected by people for its high medicinal value [1]. Ginsenosides are the main active ingredients of *P. ginseng*. Ginsenosides are triterpene glycosides formed by the condensation of sugars and aglycones [2]. These ginsenosides are called Rx: “x” represents the polarity of each ginsenoside on TLC, where the more polar component is called Ra and the less polar component is called Rh [3]. Most ginsenosides are dammarane-type tetracyclic triterpene saponins. The dammarane-type ginsenosides can be hydrolyzed to produce different saponins, which are further divided into two types: protopanaxadiol-type saponins (PPD, protopanaxadiol) and protopanaxatriol saponins (PPT, protopanaxatriol) [4]. Ginsenosides come from a range of sources in various species and possess strong efficacy, a wide range of biological activities, and high medicinal value [5]. According to research, ginsenosides demonstrate good healthcare functions for cardiovascular and nervous system diseases [6]. They also have hypoglycemic [7], hypotensive [8], antitumor [9], antiaging [10], and antifatigue effects [11], in addition to contributing to immune regulation [12]. Moreover, rare ginsenosides have more medicinal value. For example, rare ginsenosides can protect cells from apoptosis, induce the lysis of leukemia cells, and have better effects in terms of inhibiting tumor cells and immune regulation [13].

In recent years, there have been increased scientific reports on the endophytes of *P. ginseng*. Endophytes are microorganisms that live in the intercellular space of plant tissues and can symbiose with plants temporarily or permanently. They mainly include bacteria, fungi, and actinomyces. Moreover, their presence does not cause significant lesions on the host [14]. Endophytes are closely related to medicinal plants and can promote the synthesis and accumulation of the active ingredients in traditional Chinese medicinal herbs [15]. The secretions of endophytes have high medical value, which can promote immune regulation and improve antifatigue ability [16]. Some endophytes can metabolize new natural products or produce the same or similar metabolites as host plants. Thus, endophytes become a crucial source of natural products [17]. According to studies, some endophytic fungi of the *P. ginseng* genus can produce ginsenosides [18] and may present a new method of obtaining ginsenosides.

In previous studies, our group isolated an endophytic fungus strain, NSJ105, capable of producing ginsenosides from wild ginseng. The strain NSJ105 was identified as *Trametes versicolor* according to the results of the phylogenetic analysis. The extract obtained by fermentation in a PDA liquid medium could produce total saponins. This study optimized the fermentation conditions of protocols with uniform design experiments and verified the protocols to maximize the production of total saponins. Then, the type and concentration of the saponins in the fermented extract were determined, which can provide a reference value for the fermentation process of obtaining a certain saponin. The cost of producing saponins by fermentation in vitro reduced costs in large-scale production in the factory [19]. Moreover, ginsenoside-producing endophytes are an eco-friendly and inexpensive alternative to ginseng plants. Thus, the biotransformation process results in inexpensive and safe products [20]. In conclusion, this experiment has a very high application value in obtaining saponin.

## 2. Materials and Methods

### 2.1. Strains and Chemicals

The strain NSJ105 used in this experiment was an endophytic fungus isolated from wild ginseng from Changbai Mountain. Fungal Genomic DNA Extraction Kits were ordered by Solarbio Science & Technology Co. (Beijing, China). Glucose and vanillin were purchased from Tianjin Standard Technology Co. (Tianjin, China). Absolute methanol, absolute ethanol, N-butanol, and concentrated sulfuric acid were supplied by Shuangshuang Chemical Co. (Yantai, China). Chromatographically pure methanol was supplied by TEDIA (Fairfield, OH, USA). Saponin standards were ordered from Master Biotechnology Co. (Chengdu, China).

### 2.2. Identification of the Strain NSJ105

The strain NSJ105 was identified by morphological observation and ITS sequence analysis. Template DNA was extracted according to the protocol recommended by the Fungi Genomic DNA Extraction Kit. ITS sequences of endophytic fungus were amplified with the universal primers ITS1 (5′-TCCGTAGGTGAACCTGCGG-3′) and ITS4 (5′-TCCTCCGCTTATTGATATGC-3′). The reaction system for PCR amplification contained 2 × PCR MIX 25 μL, ddH2O 15 μL, DNA Template 5 μL, ITS1 2.5 μL, and ITS4 2.5 μL in a final volume of 50 μL. The thermal cycling was carried out with the following protocol: 5 min at 95 °C, followed by 35 cycles of 45 s denaturation at 94 °C, 45 s annealing at 52 °C, and 90 s extension at 72 °C, and a final extension of 10 min at 72 °C. The PCR product was separated on 1% agarose gel using electrophoresis, before sending it to BGI (Beijing, China) Co. Ltd. for sequencing. Sequence similarity was determined using NCBI Blast (https://blast.ncbi.nlm.nih.gov/Blast.cgi, accessed on 17 March 20203) and EzTaxon (https://www.ezbiocloud.net/, accessed on 18 March 2023).

Phylogenetic analysis was performed using MEGA [21] software version 5.0 with distance options according to the neighbor joining and supported with bootstrap values based on 1000 replications [22].

### 2.3. Liquid Culture of Endophytic Fungus

First, the endophytic fungus NSJ105 was inoculated on potato dextrose agar (PDA) Petri dishes and kept at 4 °C. The PDA liquid medium used for NSJ105 fermentation in vitro was created as follows: 1 L of deionized water and 200 g of diced potatoes were put into an electric cooker and boiled for 30 min. The solids were filtered out through four layers of gauze. Twenty grams of glucose was added to the remaining liquid and brought up to a volume of 1 L with deionized water, with a natural pH value. The liquid fermentation medium of 40 mL was put into a 100 mL conical flask and autoclaved at 121 °C and 0.1 MPa for 20 min using a vertical pressure steam sterilizer LDZX-50KBS (Shenan, Shanghai, China).

A 5 mm puncher was used to punch out multiple plaques on the NSJ105 PDA Petri dish, and five complete plaques (0.5 cm × 0.5 cm) were selected and inoculated into a 100 mL conical flask. Three conical flasks were inoculated as parallel experiments, and one conical flask without inoculation was reserved as a blank control. The fermentation culture was carried out with a full-temperature shaking incubator HZP-150 (Jinghong, Shanghai, China) at 150 rpm for 7 d at 26 °C. The fermentation was stopped when mycelium pellets were full of the fermentation liquid. The final growth state of mycelium pellets was photographed. Then, the whole liquid (including mycelium pellets) was dried thoroughly by an electric blast dryer DHG-9070 (Yuying, Shanghai, China) and ground evenly for total saponin extraction.

### 2.4. Extraction of Total Saponins

As seen in Figure 1, the thoroughly dried supernatant and mycelium pellets were mixed and thoroughly ground as the fermented sample to extract total saponins. First, 4 mL of anhydrous methanol and 0.2 g of the fermented sample were added to a 20 mL pellucid glass reagent bottle with a tightened screw cap. The glass reagent bottle was put into an ultrasonic instrument KM-700DE (Kunshanmeimei, Shanghai, China) at 500 W and 40 kHz for 30 min to extract total saponins. The mixture in the bottle was poured into a centrifuge tube and centrifuged by a high-speed refrigerated centrifuge KDC-160HR (Zhongjia, China) at 10,000 rpm for 10 min. The supernatant was collected. The same operation was repeated twice for the precipitation. Then, the supernatant collected thrice was evaporated using a rotary evaporation instrument RE-52AA (Xiande, Shanghai, China) to dryness at 40 °C and extracted three times by adding 2 mL of water-saturated n-butanol. Next, the supernatant was added to a rotary evaporation flask and evaporated at 60 °C until dryness. Finally, the residue was redissolved with a small amount of methanol and diluted in a 2 mL solution. The solution was filtered through a 0.22 μm syringe filter into a brown reagent bottle to obtain total saponins and stored at −20 °C.

### 2.5. Determination of Total Saponins

Determination of total saponins was carried out using the ultraviolet spectrophotometry method [18]. The method was based on a color reaction of the acid-hydrolysis products of the saponins (i.e., sapogenins) with vanillin. The concentration of total saponins (mg/mL) in the reaction sample was detected using a spectrophotometer UV-2000 (Unico, Shanghai, China) at 544 nm against a calibration curve established with an oleanolic acid (OA) standard (National Institutes for Food and Drug Control, Beijing, China).

Next, 20 μL of the total saponin solution was placed into a stoppered graduated test tube and evaporated to dryness. The residue was dissolved in 5 mL of 72% sulfuric acid mixed with 0.5 mL of 8% vanillin ethanol. The stopper test tubes were placed in a constant temperature water bath pot HH-2 (Huapuda, China) at 60 °C for 10 min and quickly cooled in an ice water bath for 10 min. The absorbance was measured at 544 nm wavelength with the reaction solution without samples as blank control. The concentration of total saponins was calculated from the standard curve.

### 2.6. Design of Fermentation Conditions

The uniform design method is widely employed to optimize experimental design. It is based on the principles of Latin square design and orthogonal design. Orthogonal design involves selecting test points from a comprehensive experiment that exhibit two key characteristics: uniform dispersion and neat comparability. “Uniform dispersion” ensures that the test points are representative, while “neat comparability” facilitates data analysis for the entire experiment. However, to achieve “neat comparability”, the test points cannot be uniformly dispersed completely, necessitating an increased number of test points. The purpose of the uniform design method is to eliminate the requirement of strict orderliness and comparability. It aims to enhance the uniformity and dispersion of test points, thereby improving their representativeness and obtaining more information with fewer tests [23]. One of the key reasons why the uniform design method is feasible is its ability to eliminate interference factors. Distributing experimental conditions evenly across different groups minimizes group differences and ensures result accuracy and reliability. It allows researchers to observe variable impacts more accurately and draw precise conclusions. It finds extensive applications in various fields and can improve efficiency and reliability for results [24,25,26].

In this experiment, potato concentration, glucose concentration, inoculation volume, fermentation broth pH, incubation temperature, and incubation time were determined as the factors of the uniform design experiment. Software Data Processing System (DPS) version 9.50 [27] was used to carry out a uniform design of mixing levels for the above six fermentation conditions with the N1–N12 protocols shown in Table 1.

### 2.7. Regression Analysis and Verification of Fermentation Protocols

Based on the concentration of total saponins determined by the above optimal fermentation protocol, DPS software was used to carry out fitting analysis on multifactor and interaction term stepwise regression, multifactor and square term stepwise regression, and quadratic polynomial stepwise regression. The optimized fermentation conditions were used to validate the regression model.

### 2.8. Identification and Analysis of Fermentation Extracts

Liquid Chromatography–Mass Spectrometry LTQ Orbitrap XL (LC–MS, Thermo Fisher Scientific, Shanghai, China) was used to determine the composition of the total saponins. Saponins were detected using an Analytical Column Accucore C18 (2.6 μm, 150 mm × 2.1 mm). Mobile phase A was deionized water. Stationary phase B was acetonitrile. Table 2 shows the elution process. The flow rate was 0.3 mL/min. The injected volume was 10 μL. The sample was detected by absorbance at 203 nm. Mass spectrum parameter was a spray voltage of 3.0 kv. The sheath gas was N_2_. Positive ion mode was adopted. The scanning range was M/Z 150–1300.

A mixed saponin standard solution with a concentration of 0.2 mg/mL was prepared, which consisted of the pseudo-ginsenoside F11, noto-ginsenoside Fe, noto-ginsenoside Ft_1_, and ginsenosides Rb_1_, Rb_2_, Rb_3_, Rc, Rd, Re, Rf, Rg_1_, Rg_2_, Rg_3_, Rh_1_, Rh_2_, Rk_2_, CK, F1, and F2. According to the total ion chromatogram (TIC) and mass spectrum (MS) analysis of each ginsenoside in the mixed saponin standards solution, the mass-to-charge ratio M/Z, peak time, and peak area of each substance were compared and analyzed with the database to determine the type and concentration of saponins [28].

The method for quantitatively calculating a certain saponin is as follows: a standard sample composed of 20 kinds of saponins and the sample to be tested (105-DP) were detected by LC–MS. The calculation for the concentration of a certain saponin is based on comparing the peak area of the saponins in the sample to be tested with the same saponins in the standard sample. The calculation formula was described using Equation (1).
(1)$$\mathrm{CST}=\frac{\mathrm{PAST}}{\mathrm{PASS}}\times \frac{\mathrm{IVT}}{\mathrm{IVS}}\times C\mathrm{SS}$$
where CST represents the concentration of a saponin in the sample to be tested; PAST represents the peak area of a saponin in the sample to be tested; PASS represents the peak area of a saponin in the standard sample; IVT represents the injection volume of the sample to be tested; IVS represents the injection volume of the standard sample; and CSS represents the concentration of a saponin in the standard sample, respectively.

In this experiment, the injection volume of the sample to be tested and the standard sample were both 4 μL. The concentration of 20 kinds of saponins in the standard sample was 0.2 mg/mL.

### 2.9. Statistical Analysis

All experiments were repeated at least thrice. Data were presented as mean ± standard deviation. Student’s t-test was used for comparison between two groups, and a one-way analysis of variance was used for comparisons among multiple groups. In the figure, * *p* < 0.05, ** *p* < 0.01, and *** *p* < 0.001 show statistically significant differences.

## 3. Results

### 3.1. Identification of the Strain NSJ105

The strain NSJ105 was an endophytic fungus isolated from wild ginseng. Its morphological characteristic shown in Figure 2 was a white filamentous fungus. NSJ105 was dense with white fluff with neat edges on the colony.

Appendix A shows that all branch confidences on the phylogenetic tree were greater than 70%. Most branch confidences were greater than 90%. This indicates that each branch of this phylogenetic tree was relatively stable and could more accurately reflect the genetic evolution relationship between various species. This result demonstrated that the strain NSJ105 and *Trametes versicolor X-02* grouped with a high branch confidence (90%). Therefore, the strain NSJ105 was designated as *Trametes versicolor* (gene accession number: OR144428).

### 3.2. Liquid Fermentation In Vitro of NSJ105

NSJ105 was inoculated into the PDA liquid medium and repeated three times. Based on Figure 3, the growth state of the fermentation liquid showed that white mycelium pellets were full of the liquid medium at the end of fermentation.

The oleanolic acid standard solution was prepared and detected for the standard curve shown in Appendix A. The correlation coefficients R^2^ (0.9973) exhibited good linearity. The optical density (OD) value at 544 nm of the fermentation extraction sample was measured, and then the total saponin concentration was calculated based on the standard curve. The extract of the NSJ105 fermentation was extracted and detected according to the ultraviolet spectrophotometry method. As depicted in Figure 4, the average total saponin concentration of the NSJ105 fermentation extract was 0.587 ± 0.039 mg/mL.

### 3.3. Total Saponins Concentration by Uniform Design Test

According to the uniform design protocols of N1–N12, the liquid fermentation of NSJ105 was carried out with each group and repeated three times. Figure 5 shows the results of the total saponin concentration of the fermentation extracts assessed by N1–N12. Compared with the total saponin concentration (0.587 mg/mL) of the original fermentation, they changed after the liquid fermentation of the strain NSJ105 by the N1–N12 uniform design protocols. The N2, N3, N4, N5, N10, and N11 protocols had lower total saponin concentration than the original protocol, while the remaining six protocols had increased total saponin concentration. According to the N6, N7, N8, and N9 protocols, the total saponin concentration was more than 1 mg/mL and nearly doubled. The N9 protocol was the best fermentation protocol as its total saponin concentration reached 1.670 mg/mL. Therefore, this protocol has the potential for optimal fitting.

### 3.4. Optimization of Fermentation Conditions

Through the regression analysis of the above uniform test results, the approximation of the optimized conditions was taken as the corresponding and achievable conditions in the practical tests of each group. The combinations with a better fit of the regression equation were selected as the optimal fermentation protocols by regression analysis, and we carried out fermentation verification. The specific protocols obtained are shown in Table 3.

The regression equation of the multifactor and interaction term stepwise regression analysis was
Y = 18.14376733 − 0.03803161354 × X1 − 2.4470010089 × X4 − 0.5960633118 × X5 − 0.00012497422046 × X1 × X2 + 0.0015899834066 × X1 × X5 + 0.00003688882864 × X1 × X6 + 0.003406528664 × X2 × X3 + 0.08327232440 × X4 × X5 + 0.04474096450 × X4 × X6−0.012526921873 × X5 × X6.(2)

The regression determination coefficient of this equation was R^2^ = 0.999975, and the significance coefficient was *p* = 0.0123. These results indicated that the equation could fit the effects of various factors on the concentration of the total saponins in fermentation extracts well.

The regression equation of the multifactor and square term stepwise regression analysis was
Y = 7.88160335 − 0.03203252539 × X1 + 0.10420532907 × X2 − 0.6162630069 × X3 − 0.6950609515 × X6 + 0.00007308741946 × X1 × X1 − 0.0026427309251 × X2 × X2 + 0.05700298596 × X3 × X3 − 0.004253341794 × X4 × X4 + 0.03165116077 × X6 × X6.(3)

The regression determination coefficient of this equation was R^2^ = 0.993211, and the significance coefficient was *p* = 0.0302. These results proved that the equation could fit the effects of various factors on the concentration of the total saponins in fermentation extracts well.

The regression equation of the quadratic polynomial stepwise regression analysis was
Y = 2.305048594 − 0.0012924897225 × X1 × X3 + 0.0019141134443 × X1 × X4 + 0.00019763880360 × X1 × X5 − 0.0012977145813 × X1 × X6 + 0.030805950287 × X2 × X3 − 0.030396877498 × X2 × X4 − 0.0016323752087 × X2 × X5 + 0.010644755631 × X2 × X6−0.0031350876937 × X3 × X5 − 0.03254371273 × X3 × X6.(4)

The regression determination coefficient of this equation was R^2^ = 0.999992, and the significance coefficient was *p* = 0.0069. These results demonstrated that the equation could fit the effects of various factors on the concentration of the total saponins in fermentation extracts well.

The above results showed that the three protocols were all combinations with a better fit of the regression equation. Thus, the three protocols could be used for optimization protocols of verification experiments.

Figure 6 shows the results of the total saponin concentration achieved by the three protocols. Compared with the predicted concentration, the total saponin concentration of the extracts after fermentation in the multifactor and interaction term stepwise regression protocol (105-DH) showed a large gap from the predicted value. The total saponin concentrations of the multifactor and square term stepwise regression (105-DP) protocol and quadratic polynomial stepwise regression protocol (105-ED) were close to the predicted value. The total saponin concentration (2.365 mg/mL) of the 105-DP protocol was the highest of the three protocols and very close to the predicted value (2.485 mg/mL), and the total saponin concentration of the 105-DP protocol was much higher than that of the original (0.587 mg/mL) and N9 protocol (1.670 mg/mL). Therefore, the 105-DP optimization protocol was the best fermentation protocol for NSJ105 to produce saponins. Its fermentation conditions were as follows: potato concentration in the liquid fermentation medium was 97.3 mg/mL, glucose concentration in the liquid fermentation medium was 20.6 mg/mL, inoculation volume was 2.1%, pH of fermentation broth was 2.1, incubation temperature was 29.2 °C, and incubation time was 6 d.

### 3.5. Identification and Analysis of Fermentation Products

The saponin standards were determined by the LC–MS method. The determination and analysis results are shown in Figure 7 and Table 4. Based on Figure 7, all the saponin standards could be detected within 30 min. The peaks were complete. Then, the 105-DP was detected by the same method as the saponin standards. The results are shown in Figure 8 and Table 5. Ginsenosides Rf and Rb_3_ were found in the fermentation extract of 105-DP. According to the calculation of their respective peak areas and total peak areas, the concentrations of the two kinds of ginsenosides were 0.66 and 0.19 mg/L. The concentration of the ginsenoside Rf was higher than Rb_3_.

## 4. Discussion

Due to the symbiotic relationship between endophytes and their hosts, endophytes impact the processes of production or accumulation of metabolites in the host. This effect is generally positive. The endophytes directly produce the same or similar metabolites as the host or promote the production or accumulation of metabolites in the host [29]. Thus, they can increase the concentration of some active metabolites. Therefore, the endophytic fungus isolated from *P. ginseng* could be fermented in vitro to produce a certain amount of total saponins. It was first found that the strain NSJ105 identified as *Trametes versicolor* could produce saponins in our study, indicating that *Trametes versicolor* could be a new fungi resource with the potential ability for saponin production. To our knowledge, there are few reports on obtaining ginsenosides in vitro using *Trametes versicolor* endophytic fungi, and the concentration of the total saponins produced by other endophytes has been reported to be relatively low, generally not exceeding 0.2 mg/mL. For example, Wu [18] found that the highest concentration of total saponins fermented in vitro by the endophytic fungus strain Pg27 was 0.181 mg/mL. Yan [30] found that the concentrations of total ginsenosides produced by the strains LB-5, PDA-2, and R2A-7 by three endophytic bacteria were 0.146, 0.073, and 0.104 mg/L, respectively, while the concentration of the total saponins (0.587 mg/mL) produced by the endophytic fungus strain NSJ105 in this research was higher than that in the above endophytes. Therefore, NSJ105 is an excellent endophytic fungus strain for fermenting and producing saponins in vitro.

The fermentation conditions of NSJ105 were optimized using the uniform design method so that the strain NSJ105 could produce the maximum total saponins. The uniform design has the advantage of time and can cover more factors and levels but lacks the superiority of accuracy [26]. In this study, the concentration of the total saponins measured in the optimal fermentation protocol 105-DP was very close to the predicted value. Compared with protocol N9 by uniform design, the 105-DP protocol reduced the concentration of potato and glucose in the medium. In this way, the later mass production process could reduce the cost, shorten the fermentation time, and improve the fermentation efficiency. Therefore, the optimization protocol 105-DP designed in this study has obvious practicability and feasibility. The optimized protocols of the fermentation conditions improved the yield of total saponins to lay the foundation for the efficient fermentation process in the later stage. However, 100 mL conical flasks were used in this study for shake flask fermentation, which had certain limitations. Therefore, we should also scale up fermentation in subsequent experiments.

Detecting the types of saponins in fermentation cultures is usually achieved via the HPLC (High-Performance Liquid Chromatography) method [18,30]. In contrast, the LC–MS method used for detecting the types and concentrations of saponins in this study is simple, fast, sensitive, and specific. The ginsenosides Rf and Rb_3_ were detected in the fermented extract of 105-DP. It was reported that ginsenoside Rb_3_ possesses the potential for clinical use in preventing and treating diabetes [31]. Ginsenoside Rb_3_ was a promising candidate for a novel class of anti-ischemic agents [32]. In addition, Ginsenoside Rb_3_ has a neuroprotective effect on memory [33]. Ginsenoside Rf is a tetracyclic triterpenoid unique to *P. ginseng*. It has a wide of effects on the nervous system, digestive system, and cardiovascular system [34]. The ginsenoside Rf could be transformed into the rare ginsenoside Rh_1_, and the rare ginsenoside Rh_1_ showed high cytotoxicity against tumor cells [35]. Moreover, the ginsenoside Rf could be transformed into 20(S)-protopanaxatriol (PPT(S)) using glycosidase from *Aspergillus niger*. However, the preparation of PPT(S) is very difficult due to the natural absence of PPT(S) in *P. ginseng*. Therefore, it is desirable to prepare PPT(S) via the biotransformation of ginsenoside Rf [36]. In conclusion, the endophytic fungus strain NSJ105 could be used as a direct source of the ginsenosides Rf and Rb_3_ in the pharmaceutical industry and a crucial strain for the indirect transformation of the rare ginsenosides Rh_1_ and PPT(S).

To sum up, ginsenosides are obtained by chemically extracting *P. ginseng* through long-term cultivation. The process is time-consuming and costly, and the extraction process is not conducive to environmental protection. A novel method using an endophytic fungus to produce ginsenosides was proposed to solve this problem. The superiority of the naturally produced active substances by endophytic fungus is obvious. Endophytic fungi not only grow rapidly but are also easily cultivated as a result of no affectation by environmental factors. In addition, the endophytic fungi can be easily screened for high-yield strains utilizing mutation breeding to increase yield, and microbial fermentation is greatly beneficial in industrial production for its low cost. Therefore, the results of this study could be significant in the production of ginsenosides.

## 5. Conclusions

The *Trametes versicolor* endophytic fungus NSJ105 isolated from wild ginseng can produce saponins through in vitro fermentation. The optimal fermentation protocol 105-DP was as follows: potato concentration 97.3 mg/mL, glucose concentration 20.6 mg/mL, inoculum 2.1%, fermentation broth pH 2.1, fermentation temperature 29.2 °C, and fermentation time 6 d. The 105-DP protocol can produce 2.365 mg/mL of total saponins and was at least three times higher than the original protocol (0.587 mg/mL). It contained the ginsenosides Rf and Rb_3_. There are few reports about the endophytic fungi *Trametes versicolor* being capable of producing saponins. The strain NSJ105 has potential application value in saponin production.

## Figures and Tables

**Figure 1 microorganisms-11-02331-f001:**
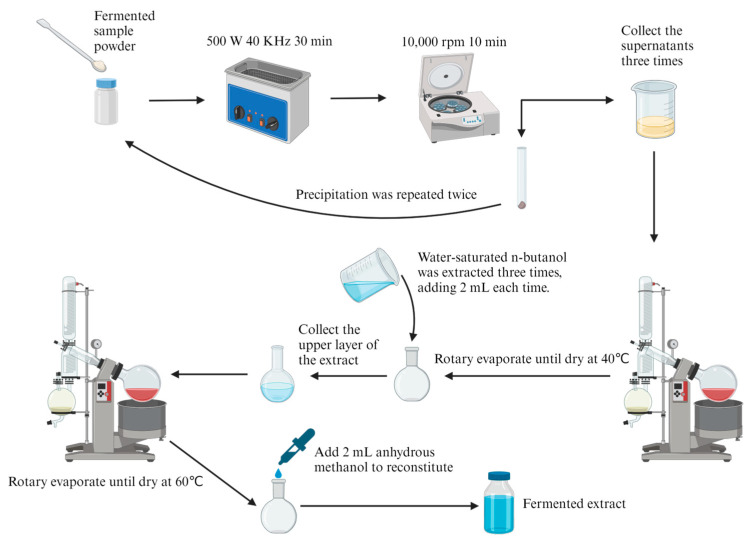
The extraction process of total saponins from fermented sample.

**Figure 2 microorganisms-11-02331-f002:**
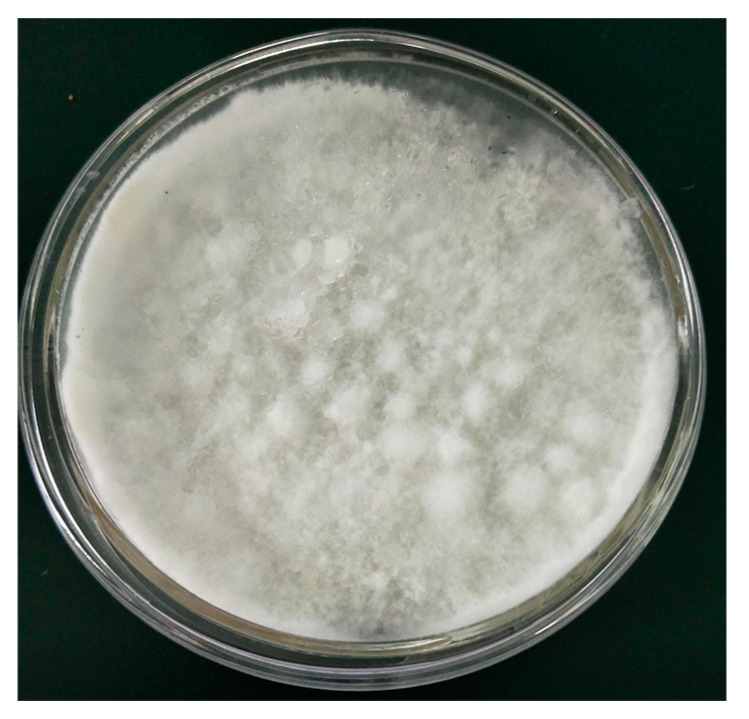
Morphological characteristic of the strain NSJ105.

**Figure 3 microorganisms-11-02331-f003:**
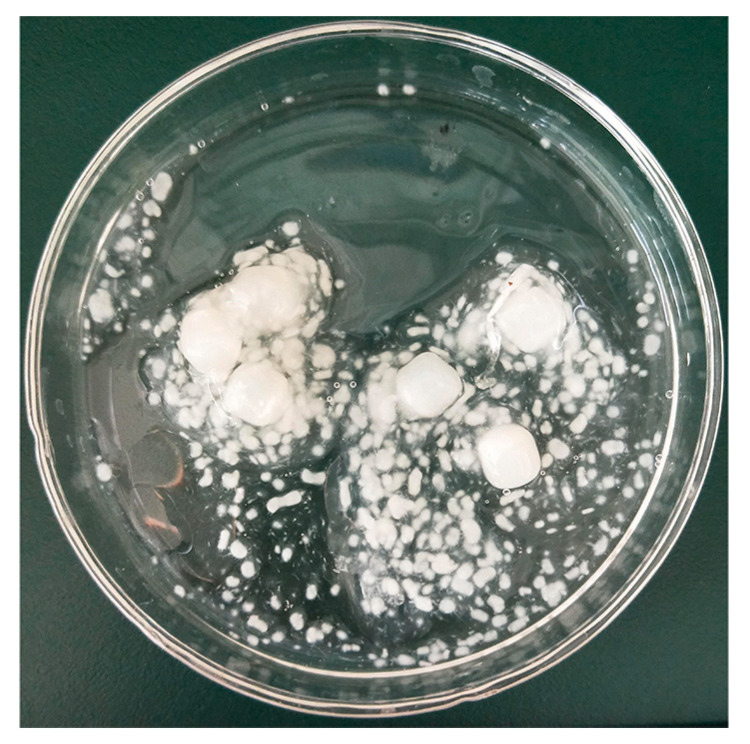
Growth state of the fermentation liquid by the strain NSJ105.

**Figure 4 microorganisms-11-02331-f004:**
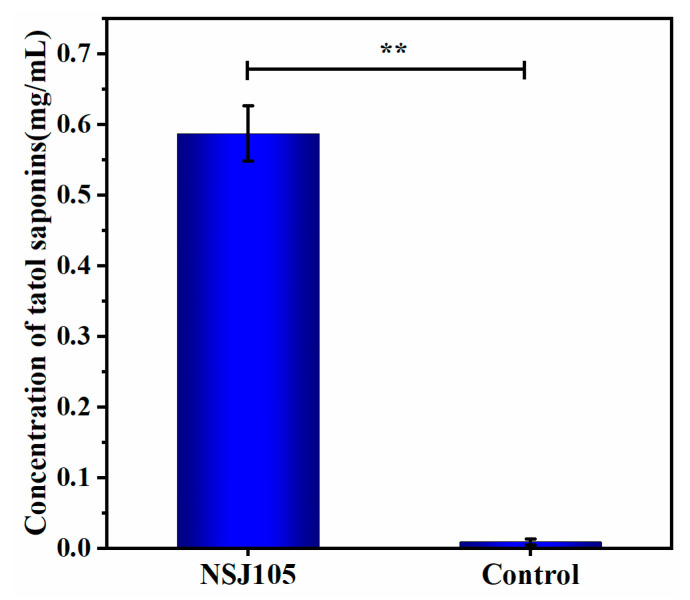
The concentration of the original protocol of PDA fermented by the strain NSJ105. Data are expressed as mean ± SD (*n* = 3). ** *p* < 0.01.

**Figure 5 microorganisms-11-02331-f005:**
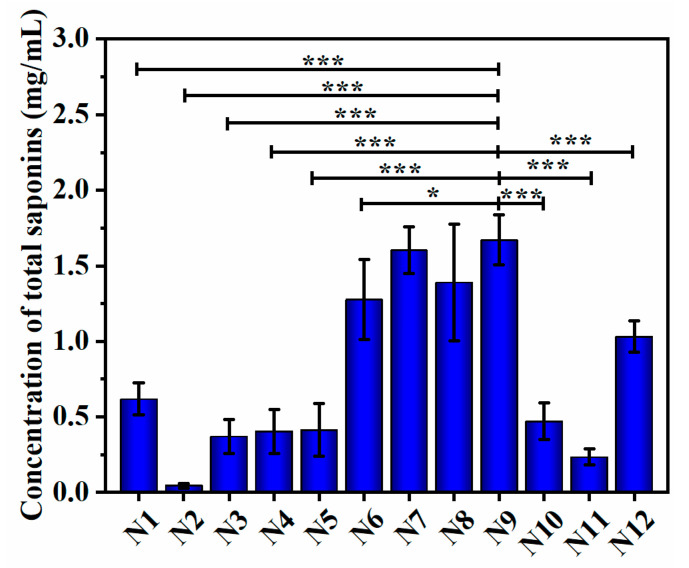
Concentration of total saponins in fermentation extracts by 12 uniform design protocols. N1–N12 protocols containing the six mixing levels (potato concentration, glucose concentration, inoculation volume, fermentation broth pH, incubation temperature, and incubation) in the uniform design experiment were designed using DPS software version 9.0. The concentration of total saponins by N1–N12 protocols was determined by the ultraviolet spectrophotometry method. Data are expressed as mean ± SD (*n* = 3). * *p* < 0.05, *** *p* < 0.001.

**Figure 6 microorganisms-11-02331-f006:**
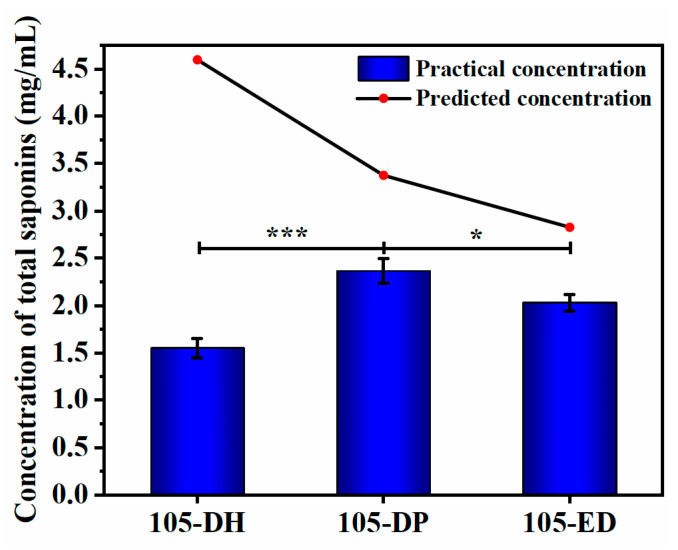
Concentration of total saponins in fermentation extracts by three optimization protocols. The optimal protocols 105-DH, 105-DP, and 105-ED for fermentation verification were obtained by multifactor and interaction term stepwise regression analysis, multifactor and square term stepwise regression analysis, and quadratic polynomial stepwise regression analysis. The predicted concentration of total saponins was calculated by DPS software 9.0. The practical concentration values are presented as means ± SD (*n* = 3). * *p* < 0.05, *** *p* < 0.001.

**Figure 7 microorganisms-11-02331-f007:**
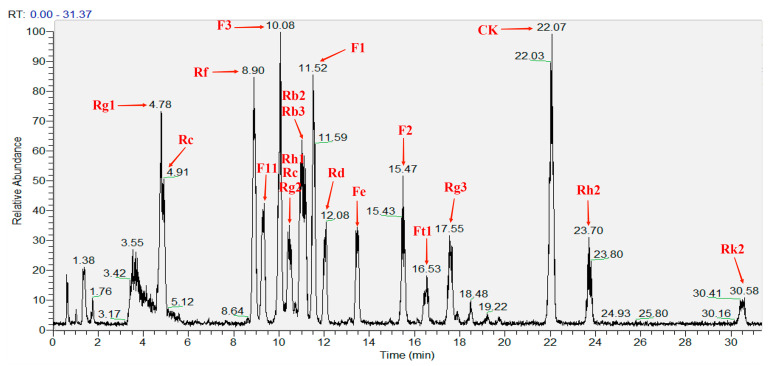
TIC of LC–MS determination by 19 saponin standards.

**Figure 8 microorganisms-11-02331-f008:**
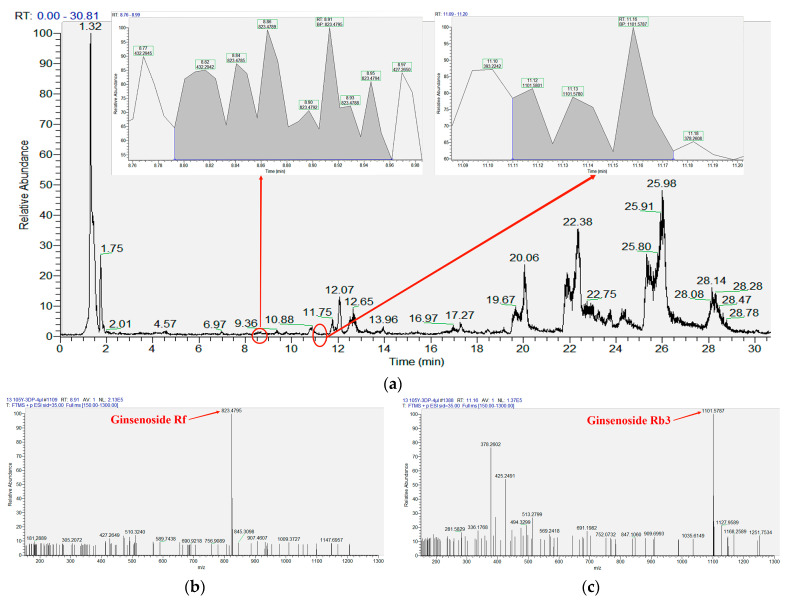
Determination and analysis of saponins. (**a**) TIC of LC–MS determination by 105-DP fermentation extract including partial enlargement of ginsenoside Rf and Rb_3_. (**b**) MS of ginsenoside Rf was 823.4795. (**c**) MS of ginsenoside Rb_3_ was 1101.5787.

**Table 1 microorganisms-11-02331-t001:** Uniform design protocols of mixing level.

Factors	Potato Concentration (g/L)	Glucose Concentration (g/L)	Inoculation Volume (%)	pH	Incubation Temperature (°C)	Incubation Time (d)
N1	100	10	4	6.0	28	10
N2	200	35	4	7.0	22	6
N3	200	15	2	7.0	22	12
N4	200	35	2	5.0	28	10
N5	300	25	4	4.0	28	12
N6	100	30	2	4.0	22	8
N7	300	20	2	6.0	28	6
N8	100	25	8	7.0	28	8
N9	100	20	8	5.0	22	12
N10	300	10	4	5.0	22	8
N11	300	30	8	6.0	22	10
N12	200	15	8	4.0	28	6

**Table 2 microorganisms-11-02331-t002:** LC–MS gradient elution procedure.

Time (min)	Water (%)	Acetonitrile (%)
0	77	23
13	54	46
33	32	68
45	32	68
55	0	100
60	0	100
63	77	23

**Table 3 microorganisms-11-02331-t003:** The optimization protocols for multifactor regression analysis and fitting.

Protocols	Multifactor and Interaction Term Stepwise Regression (105-DH)	Multifactor and Square Term Stepwise Regression (105-DP)	Quadratic Polynomial Stepwise Regression (105-ED)
Potato concentration (g/L)	103.5	97.3	202.3
Glucose concentration (g/L)	35.5	20.6	10.6
Inoculation (%)	8.2	2.1	2.0
pH	2.0	2.1	7.7
Incubation temperature (°C)	21.2	29.2	28.8
Incubation time (d)	6	6	6
Predicted concentration (mg/mL)	3.385	2.485	2.082

**Table 4 microorganisms-11-02331-t004:** Results of LC–MS analysis of 19 saponin standards.

Standard Name	RT (min)	Area	M/Z (M + Na)	Calculate Mass Errors (ppm)
Ginsenoside Rg_1_	4.78	210,621,828	823.4778	3.76
Ginsenoside Re	4.91	185,905,150	969.5354	3.51
Ginsenoside Rf	8.90	326,583,879	823.4778	3.76
Pseudo-ginsenoside F11	9.34	193,324,327	823.4777	3.89
Ginsenoside F3	10.08	359,421,580	793.4682	2.65
Ginsenoside Rh_1_	10.45	33,692,890	661.4218	9.37
Ginsenoside Rc	10.52	56,417,603	1101.5771	3.54
Ginsenoside Rg_2_	10.72	39,293,309	807.4778	10.03
Ginsenoside Rb_2_	11.01	223,210,191	1101.5769	3.72
Ginsenoside Rb_3_	11.11	189,872,269	1101.5764	4.18
Ginsenoside F1	11.52	313,614,257	661.4257	3.48
Ginsenoside Rd	12.08	158,247,318	969.5354	3.51
Noto-ginsenoside Fe	13.47	159,923,021	939.5243	4.15
Ginsenoside F2	15.47	185,211,392	807.4832	3.34
Noto-ginsenoside Ft_1_	16.53	81,671,445	939.5245	3.94
Ginsenoside Rg_3_	17.55	186,817,422	807.4833	3.22
Ginsenoside CK	22.07	466,725,112	645.4313	2.79
Ginsenoside Rh_2_	23.70	115,834,711	645.4308	3.56
Ginsenoside Rk_2_	30.58	65,261,096	627.4207	3.03

**Table 5 microorganisms-11-02331-t005:** Results of LC–MS analysis of 105-DP fermentation extract.

Compound Name	RT (min)	M/Z (M + Na)	Calculate Mass Errors (ppm)	Area	Concentration (mg/L)
Ginsenoside Rf	8.91	823.4795	1.70	2,145,591	0.66
Ginsenoside Rb_3_	11.16	1101.5787	2.09	352,600	0.19

## Data Availability

The data of all results in this study are included in the manuscript. If necessary, the data can be obtained by contacting the corresponding author.

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
