# Peer review of "Optimization of Fermentation Conditions and Product Identification of a Saponin-Producing Endophytic Fungus"

_microorganisms, 2023, doi:10.3390/microorganisms11092331_

Round 1
Reviewer 1 Report
The manuscript titled "Optimization of Fermentation Conditions and Product Identification of a Saponin-Producing Endophytic Fungus" aims to optimize fermentation parameters for saponin production by Trametes versicolor. The manuscript needs to be proofread in English before it can be properly reviewed. Additionally, the quality of the figures needs to be improved.
Majors
1. The standard curves and phylogenetic tree are suggested to go to the supplementary files.
2. Statistical analysis should be performed.
3. It is important to clearly identify the saponins produced during fermentation. The authors failed to identify the produced saponins, as shown in Figures 7 and 8, and did not describe any increase in saponin production. Additionally, the authors need to provide a detailed description of how they quantify the saponins.
4. Uniform design literature should be added, and its reliability should be described.
Minors
1. The authors were encouraged to re-phrase the abstract, which is a bit redundant.
2. The literatures regarding producing saponins from Trametes should be added into introduction.
The manuscript needs to be proofread in English before it can be properly reviewed.
Reviewer 2 Report
Optimization of fermentation conditions and product identification of a saponin-producing endophytic fungus
1- Introduction
Therefore, the cost of producing saponins by fermentation in vitro was lower than cultivating P. ginseng and extracting their saponins. Moreover, the process of fermentation was environmentally friendly because there were no excess metabolites in the fermentation process.
Some bibliographical references that support this paragraph would be appropriate.
2- 2.3.2. Liquid culture of endophytic fungus
This experimental section should be expanded, more details of the process would be appropriate. Some specifications of the equipment used should be mentioned.
3- 2.3.3. Extraction of total saponins
Include a representative figure of the extraction process
After the suggested comments, the manuscript should be considered for acceptance.
Some minor editing of English language required
